# Dynamic Ring Exploration with $(H, S)$ View [†]

**Tsuyoshi Gotoh [1],\*, Yuichi Sudo [1], Fukuhito Ooshita [2] and Toshimitsu Masuzawa [1]** 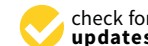

[1]   Graduate School of Information Science and Technology, Osaka University, 1-5 Yamadaoka, Suita, Osaka 565-0871, Japan; y-sudou@ist.osaka-u.ac.jp (Y.S.); masuzawa@ist.osaka-u.ac.jp (T.M.)

[2]   Nara Institute of Science and Technology, 8916-5 Takayamacho, Ikoma, Nara 630-0101, Japan; f-oosita@is.naist.jp

\*   Correspondence: t-gotoh@ist.osaka-u.ac.jp

[†]   This paper is an extended version of our paper published in the proceedings of the 21st International Symposium on Stabilization, Safety, and Security of Distributed Systems (SSS 2019), Pisa, Italy, 22–25 October 2019.

**Abstract:** The researches about a mobile entity (called agent) on dynamic networks have attracted a lot of attention in recent years. Exploration which requires an agent to visit all the nodes in the network is one of the most fundamental problems. While the exploration of dynamic networks with complete information or with no information about network changes has been studied, an agent with partial information about the network changes has not been considered yet despite its practical importance. In this paper, we consider the exploration of dynamic networks by a single agent with partial information about network changes. To the best of our knowledge, this is the very first work to investigate the exploration problem with such partial information. As a first step in this research direction, we focus on 1-interval connected rings as dynamic networks in this paper. We assume that the single agent has partial information called the $(H, S)$ view by which it always knows whether or not each of the links within $H$ hops is available in each of the next $S$ time steps. In this setting, we show that $H + S \geq n$ and $S \geq \lceil n/2 \rceil$ ($n$ is the size of the network) are necessary and sufficient conditions to explore 1-interval connected rings. Moreover, we investigate the upper and lower bounds of the exploration time. It is proven that the exploration time is $O(n^2)$ for $\lceil n/2 \rceil \leq S < 2H' - 1$, $O(n^2/H + nH)$ for $S \geq \max(\lceil n/2 \rceil, 2H' - 1)$, $O(n^2/H + n \log H)$ for $S \geq n - 1$, and $\Omega(n^2/H)$ for any $S$ where $H' = \min(H, \lfloor n/2 \rfloor)$.

**Keywords:** distributed algorithms; dynamic networks; 1-interval connected rings; mobile agent; exploration

## 1. Introduction

More applications of dynamic networks have arisen in recent years, for example, wireless mobile ad hoc, transportation, inter vehicle, or social networks and so on, more important the researches about the dynamic networks have got. A network is *dynamic* if its topology may change with time (due to various reasons, e.g., faults or movements of nodes). In a dynamic network, existing methods designed for static networks (the topologies of which do not change with time) might no longer work. For this reason, the researchers have started to consider several problems on dynamic networks [1].

The exploration which requires a mobile entity called an agent (e.g., a software agent, a robot, or a vehicle) to visit all the nodes of the network is one of the most fundamental problems. The exploration is useful for solving fundamental tasks on the networks such as broadcast or network maintenance. It has been well-studied for static networks [2] and recently been studied for dynamic networks. In the previous works about the exploration of dynamic networks, two extreme cases are considered: an agent has the a priori complete knowledge about changes of all the links for all the future time steps [3–7]; or an agent can only see whether the links adjacent to its current node are present or

not at the moment [8–12]. The former one models the situation where the network changes are completely predictable as the public transportation networks in which the network changes are introduced by totally scheduled movements of the nodes. The latter one models the situation where the network changes are caused by unscheduled events, for example, faults or unscheduled movements of the nodes.

Although the above two models are plausible and also theoretically important, the intermediate model, i.e., an agent with partial information or, in other words, capability to know link changes within some distance in the near future should be considered due to the following reasons: even in the totally scheduled situation (if exists), computing all the future changes often costs computation time and it is desirable to compute only the necessary information to solve a problem to save computing time or memories; the ability of an agent to monitor whether there are faults or environmental changes roughly depends on the quality (or costs) of its sensor and it can save some costs to compute only the necessary information for a problem. Moreover, such a model is so interesting from a theoretical viewpoint: how the amount of information available for an agent influences the solvability or the time complexity of problems.

In this paper, we consider the exploration of dynamic networks by a single agent with partial information about network changes. To the best of our knowledge, this is the very first work to investigate the exploration using such partial information. As a first step in this research direction, we focus on 1-interval connected rings as dynamic networks in this paper. To formalize the concept of partial information and analyze its influences, in this paper, we first propose the $(H, S)$ *view* such that the agent with the view can see the *link scheduling* (when and which links disappear or appear) of the links within $H$ hops from its location for $S$ time steps from the current time. Then, we consider how the value of $H$ or $S$ influences the solvability or the time complexity of the exploration by a single agent of 1-*interval connected rings* in which at most one link is missing at each time step. While the 1-interval connected rings are probably too restrictive from a practical point of view, they are adequate targets to investigate in the novel direction as investigated in many works (e.g., in the field of mobile agents on dynamic networks, [7,10,13–15] consider 1-interval connected rings).

*1.1. Related Works*

To see various settings and exploration algorithms on static networks, there is a good survey [2], for example, a network with distinct node labeling or without node labeling (an anonymous network), exploration with termination or perpetual exploration, and from the point of the number of agents, exploration by a single agent or by multiple agents.

The literature of dynamic networks are surveyed in [1,16].

The recent works about mobile agents (or robots) on dynamic networks are summarized in [17] including exploration; *gathering* on 1-interval connected rings [13] which requires all the agents to gather at one node or at adjacent two nodes; *dispersion* [18] which stipulates that every node must be occupied by exactly one agent where the number of agents is the same as that of nodes on *permuting rings* in which the nodes may be permuted at each time step, i.e., the neighbors of a node may change at each time step while the topologies are rings or paths at each time step.

The following works consider the exploration of dynamic networks by multiple agents (or robots) without the knowledge of a link scheduling (or only with the ability to detect whether the adjacent links are present or not at the moment). In [10], the exploration for 1-interval connected rings is considered. The *perpetual exploration* (i.e., the exploration without termination) on *connected-over-time rings* is considered in [8,9]. In [11], the perpetual exploration on two kinds of temporal networks with arbitrary footprints is considered: *connected-over-time graphs* and 1-*interval connected graphs with bounded missing links*. The difference between with or without the ability to detect whether the adjacent links are present or not at the moment (called the *link presence detection*) is considered in [12] for an $n \times m$ *dynamic torus* which consists of $n$ horizontal rings and $m$ vertical rings each of which is a 1-interval

connected ring. It is shown that the minimum number of agents with the link presence detection to explore the networks is a half of the minimum number of agents without the one to explore.

The following works consider the exploration of dynamic networks by an agent with the full knowledge of a link scheduling, i.e., the information about when and which links disappear or appear. In [7], the exploration is considered on *T-interval connected rings* where at most one link is missing at each time step and for any $T$ successive time steps, there exists a common spanning connected component. It is shown that the optimal exploration time is $2n - 3$ when $T = 1$ where $n$ is the network size. In [6], the exploration on *1-interval connected cactuses* is considered. They show that the graphs can be explored in $2^{O(\sqrt{\log n})}$ time which is much less than the known upper bound for the general graph, $O(n^2)$. In [3], the authors reveal the existence of the 1-interval connected graphs which have the exploration time $\Omega(n^2)$, proving the exploration time of arbitrary 1-interval connected graphs is $\Theta(n^2)$. In [5], it is shown that when the maximum degree at each time step is upper-bounded by $d$, the exploration time is reduced to $O((dn^2 \log d) / \log n)$. In [4], the authors prove that the exploration time is reduced to $O(n^{1.75})$ if an agent can move two hops in each time step.

Other problems are also considered on dynamic networks; *patrolling* on 1-interval connected rings [14] which requires the maximum length of the interval between two visits to a node to be minimized; *compacting* on 1-interval connected rings [15] which stipulates that all the agents in a network must be located in a continuous part of the ring and at each node there exists at most one agent.

*1.2. Our Contributions*

In this paper, we consider the exploration of 1-interval connected rings by a single agent with the $(H, S)$ view (formalizing the proofs and the pseudo codes and extending the results given in [19]). Remind that the agent with the $(H, S)$ view can see the link scheduling of the links within $H$ hops $(1 \leq H \leq \lceil n/2 \rceil)$ from its location for $S$ time steps from the current time. To the best of our knowledge, this is the first work to generalize the agent capacity to see a link scheduling.

The results are summarized in Table 1. For the proposed model, we show that $H + S \geq n$ and $S \geq \lceil n/2 \rceil$ ($n$ is the size of networks) are the necessary and sufficient conditions to explore 1-interval connected rings by a single agent. We also show that in the case where the above conditions holds, the exploration can be achieved within $O(n^2)$ time if $2H' - 1 > S$ or otherwise $O(n^2/H + nH)$ time where $H' = \min(H, \lfloor n/2 \rfloor)$. This is a new addition to the contributions of the previous work [19]. Moreover, we show that when $S \geq n - 1$, the exploration time can be reduced to $O(n^2/H + n \log H)$. This leads to $O(n \log n)$ time when $H = \Theta(n/\log n)$. Finally, we show a lower bound of the exploration time, $\Omega(n^2/H)$, for any $S$. This implies that we have tight bound $\Theta(n^2/H)$ when $H + S \geq n$, $\max(\lceil n/2 \rceil, 2H' - 1) \leq S$, and $H$ is $O(n^{0.5})$ and when $S \geq n - 1$ and $H = O(n/\log n)$.

**Table 1.** Upper and lower bounds of the exploration time in 1-interval connected rings where $H' = \min(H, \lfloor n/2 \rfloor)$.

| $H$ and $S$ | Upper Bound | Lower Bound |
|:---:|:---:|:---:|
| $H + S < n$ or $S < \lceil n/2 \rceil$ | The exploration is **impossible**. | |
| $H + S \geq n$ and $\lceil n/2 \rceil \leq S < 2H' - 1$ | $O(n^2)$ | |
| $H + S \geq n$ and $\max(\lceil n/2 \rceil, 2H' - 1) \leq S < n - 1$ | $O(n^2/H + nH)$ | $\Omega(n^2/H)$ |
| $n - 1 \leq S$ | $O(n^2/H + n \log H)$ | |

## 2. Models and Terminologies

We consider *a time variant ring* $\mathcal{R} = (V, E, \rho)$ where $G = (V, E)$ is a ring network, i.e., $V = \{v_0, v_1, \ldots, v_{n-1}\}$ is a set of $n$ nodes and $E = \{e_0, e_1, \ldots, e_{n-1}\}$ is a set of $n$ links such that $e_i = (v_i, v_{i+1 \bmod n})$. The nodes of the network are anonymous. For simplicity, we omit mod $n$ in the following. A function $\rho : E \times \mathbb{N} \to \{0, 1\}$ is called a *link presence function* such that $\rho(e, t)$ is 1 (resp., 0) if link $e$ is present (resp., missing) at time step (or *step*) $t \in \mathbb{N}$. A network at each step $t$ is denoted as $R_t = (V, E_t)$ where $E_t = \{e_i \in E \mid \rho(e_i, t) = 1\}$. We assume that $\mathcal{R}$ is 1-*interval connected*, i.e., at each step $t$, a network $R_t$ is connected. In other words, at each step $t$, there is at most one missing link $e_i \in E$ such that $e_i \notin E_t$.

We say the ascending (resp., descending) order of node indices is the right (resp., left) direction. Each port of $e_i$ has a globally consistent label at $v_i$ and $v_{i+1}$ which gives an entity on the ring a global direction (the right direction at $v_i$ and the left direction at $v_{i+1}$) of the ring. Given a connected component $V' \subsetneq V$, the *right* (resp., *left*) *extremity* of $V'$ is the node $v_i \in V'$ such that $v_{i+1} \notin V'$ (resp., $v_{i-1} \notin V'$). If $|V'| = 1$, the unique node in $V'$ is both the right extremity and the left extremity of $V'$.

In the network, a single agent $A$ is operational. Agent $A$ knows the network size $n$, has computation capacity and its own memory, and can traverse at most one link in each step. In addition to them, $A$ can get the *view* which contains information of presence of nearby links in near future as defined later. In a step $t$, $A$ at a node, say $v_i$, first decides which direction it moves and updates its memory depending on the current content of its memory and the view from $v_i$. If the corresponding link is present at $t$, $A$ succeeds to move and reaches a neighbor of $v_i$ by the end of $t$. Otherwise, $A$ fails to move and stays at $v_i$.

Informally speaking, the $(H, S)$ *view* that agent $A$ can get shows which link is missing within $H$ hops from the current node and within $S$ steps in the future including the current step. Formally speaking, for $\lceil n/2 \rceil \geq H \geq 1$ and $S \geq 1$, $A$ gets the $(H, S)$ view $\beta_{H,S}(i, s) = \{(e_j, t, \rho(e_j, t)) \mid i - H \leq j \leq i + H - 1, s \leq t \leq s + S - 1\}$ when $A$ exists on $v_i$ at step $s$. For example, when $H = 2$, $S = 2$, and $A$ exists on $v_0$ at step 5, $A$ can see $\beta_{2,2}(0, 5) = \{(e_1, 5, 0), (e_0, 5, 1), (e_{n-1}, 5, 1), (e_{n-2}, 5, 1), (e_1, 6, 1), (e_0, 6, 0), (e_{n-1}, 6, 1), (e_{n-2}, 6, 1)\}$. When no confusion arises, we simply write *the view* instead of writing the "$(H, S)$ view".

It is assumed that a link scheduling (or $\rho(e_i, t)$ for every $e_i$ and every step $t > 0$) is decided by the adversary. The adversary knows the algorithm of $A$, has infinite computation capacity, and tries to prevent $A$ from exploring the ring.

In this paper, we consider the exploration problem by a single agent $A$: $A$ is required to visit all the nodes in the ring. A node is said to be *explored* by (resp., at) the $t$-th step when it is visited by $A$ at the end of the $(t - 1)$-th step or earlier (resp., at the end of the $(t - 1)$-th step for the first time). In the similar manner, we say that $A$ reaches a node at the $t$-th step when $A$ visits the node at the end of the $(t - 1)$-th step and that $A$ *explores* a node $v$ at the $t$-th step if $v$ is unexplored at the start of the $(t - 1)$-th step and $A$ reaches $v$ at the $t$-th step. The set of explored (resp., unexplored) nodes at the start of the $t$-th step is denoted by $V^t$ (resp., $\overline{V^t}$). Without loss of generality, we assume $A$ starts the exploration from $v_0$.

In the following, we use "to move to right (resp., left)" instead of "to move in the right (resp., left) direction" for simplicity.

## 3. Impossibility Result

We show an impossibility result in this section. Specifically, we show that the exploration is impossible when $H + S < n$ or $S < \lceil n/2 \rceil$ holds.

**Lemma 1.** *If $H + S < n$ or $S < \lceil n/2 \rceil$, a deterministic single agent with the $(H, S)$ view cannot explore 1-interval connected rings.*

**Proof.** We first consider the condition $S < \lceil n/2 \rceil$. We assume $H = \lceil n/2 \rceil$. It suffices to show that the exploration is impossible when $S = \lceil n/2 \rceil - 1$. We assume for contradiction, that there is an algorithm by which $A$ can explore any ring under any link scheduling when $S = \lceil n/2 \rceil - 1$. Since $A$ can explore the ring, $A$ starting from $v_0$ eventually reaches $v_{n-1}$ (no matter whether the exploration is completed or not).

The adversary decides a link scheduling so that $e_{n-1}$ (resp., $e_{n-2}$) is missing when $A$ exists on $v_0$ (resp., $v_{n-2}$). The adversary first keeps showing a link scheduling where $e_{n-1}$ is kept deleted for $S$ steps from the current step until $A$ moves to $v_{n-\lceil n/2 \rceil}$. If $A$ does not move to $v_{n-\lceil n/2 \rceil}$ and stays $v_i$ for $0 \leq i < n - \lceil n/2 \rceil$, $e_{n-1}$ is kept deleted and $A$ cannot reach $v_{n-1}$ ($A$ must pass through $e_{n-1}$ or $e_{n-\lceil n/2 \rceil - 1}$ to reach $v_{n-1}$ from $v_0$), which is a contradiction. Thus, $A$ eventually reaches $v_{n-\lceil n/2 \rceil}$ at some step, say $t$.

Then, the adversary deletes $e_{n-2}$ from the $(t + S - 1)$-th step (the $(t + \lceil n/2 \rceil - 2)$-th step) until $A$ moves to $v_{n-\lceil n/2 \rceil - 1}$. By the scheduling, since $A$ reaches $v_{n-2}$ at earliest at the $(t + n - 2 - (n - \lceil n/2 \rceil))$-th step (the $(t + \lceil n/2 \rceil - 2)$-th step) from $v_{n-\lceil n/2 \rceil}$, $e_{n-2}$ starts to disappear when (or before) $A$ reaches $v_{n-2}$ and keeps disappearing unless $A$ moves to $v_{n-\lceil n/2 \rceil - 1}$. Thus, if $A$ does not move to $v_{n-\lceil n/2 \rceil - 1}$, $A$ cannot reaches $v_{n-1}$. This is a contradiction.

This means that $A$ moves to $v_{n-\lceil n/2 \rceil - 1}$ after the $t$-th step. However, by the similar way, the adversary can prevent $A$ from reaching $v_{n-1}$. This is a contradiction. Hence, when $S < \lceil n/2 \rceil$, a single agent cannot explore 1-interval connected rings.

Secondly, we consider the condition $H + S < n$ and $S \geq \lceil n/2 \rceil$. It is sufficient to show that $A$ cannot explore the ring when $S = n - H - 1$ for $1 \leq H \leq \lfloor n/2 \rfloor - 1$ since $H < \lfloor n/2 \rfloor$ from the conditions. Again, we assume for contradiction, that there is an algorithm by which $A$ can explore any ring under any link scheduling. Since $A$ can explore the ring, $A$ starting from $v_0$ eventually reaches $v_{n-1}$ (no matter whether the exploration is completed or not).

The adversary first keeps showing a link scheduling where $e_{n-1}$ is kept deleted for $S$ steps from the current step until $A$ moves to $v_H$. If $A$ does not move to $v_H$ and stays at $v_i$ for $0 \leq i \leq H - 1$, $e_{n-1}$ is kept deleted and $A$ cannot reach $v_{n-1}$, which is a contradiction. Thus, $A$ eventually reaches $v_H$ at some step, say $t$. After step $t$, depending on whether $A$ reaches $v_{H-1}$ before $v_{n-H-1}$ or not, the missing link is decided (Figure 1). Note that since $H \leq \lfloor n/2 \rfloor - 1$, $(n - H - 1) - (H - 1) \geq 2$ and there exists a node $v_i$ such that $H \leq i \leq n - H - 2$. Moreover, $A$ can see neither $e_{n-1}$ nor $e_{n-2}$ in its view when existing at $v_i$ for $H \leq i \leq n - H - 2$.

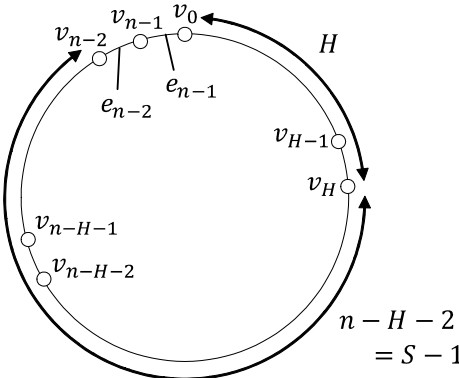

**Figure 1.** Illustrating the proof of Theorem 1 for the case of $H + S < n$ and $S \geq \lceil n/2 \rceil$.

If $A$ reaches $v_{H-1}$ before $v_{n-H-1}$, the adversary keeps deleting $e_{n-1}$. By the link scheduling, unless $A$ decides to reach $v_{n-H-1}$ from $v_H$, $e_{n-1}$ is kept deleted and $A$ cannot reach $v_{n-1}$, which is a contradiction. This means that $A$ eventually reaches $v_{n-H-1}$. Let $t'$ be the last step before $A$ reaches $v_{n-H-1}$ such that $A$ exists at $v_{H-1}$ at the start of $t'$.

When $A$ leaves $v_{H-1}$ at the $t'$-th step, the adversary makes a scheduling so that $e_{n-2}$ starts and keeps disappearing from the $(t' + n - H - 1)$-th step until $A$ comes back to $v_{n-H-2}$. This does not

conflict with the link scheduling in the past view of $A$ since at the $t'$-th step, $e_{n-1}$ is scheduled to be deleted for the next $S = n - H - 1$ steps and for the next $n - H - 1 - x$ steps at the $(t' + x)$-th step.

Since it takes at least $n - H - 2$ steps to reach $v_{n-2}$ from $v_H$, $A$ reaches $v_{n-2}$ at earliest at the $(t' + n - H - 1)$-th step. However, at the $(t' + n - H - 1)$-th step, $e_{n-2}$ is missing and the adversary keeps deleting $e_{n-2}$ until $A$ reaches $v_{n-H-2}$. Then, $A$ cannot reach $v_{n-1}$ unless moving to $v_{n-H-2}$. However, by the similar way, the adversary can prevent $A$ from reaching $v_{n-1}$. This is a contradiction. Hence, when $H + S < n$ or $S < \lceil n/2 \rceil$, a single agent cannot explore 1-interval connected rings. □

## 4. Possibility Result and Upper Bounds of Exploration Time

In this section, we prove the exploration is possible when $H + S \geq n$ and $S \geq \lceil n/2 \rceil$ by giving an exploration algorithm by a single agent. In the following, we use $H' = \min(H, \lfloor n/2 \rfloor)$. The algorithm also gives upper bounds of the exploration time, $O(n^2/H + nH)$ if $2H' - 1 \leq S$ or otherwise $O(n^2)$. Note that $S \geq H$ since $S \geq \lceil n/2 \rceil$ and $H \leq \lceil n/2 \rceil$.

We first introduce two operations $\text{EXPH}(t, v_i)$ and $\text{EXPONE}(t, v_i)$ that are used as building blocks to construct the exploration algorithm.

In the algorithms, *Extremity*$(t, v)$ is a function which returns *right* if $v$ is the right extremity of $V^t$, *left* if $v$ is the left extremity, or otherwise *nil*. Variable *dir* is used to store the direction and $\overline{dir}$ denotes the other direction (e.g., if *dir* is *right*, $\overline{dir}$ is *left*).

**EXPH.** $\text{EXPH}(t, v_i)$ described in Algorithm 1 is an algorithm by which $A$ explores $H'$ nodes when $A$ starts $\text{EXPH}(t, v_i)$ from $v_i$ at the $t$-th step under the assumption that $v_i$ is the right or left extremity of $V^t$ and $2H' + |V^t| - 1 \leq \min(S + 1, n)$. Note that in the following, when $A$ executes $\text{EXPH}(t, v_i)$, $A$ is always on the right or left extremity of $V^t$.

When starting the algorithm, $A$ first sees if $v_i$ is the right extremity or the left one and stores *right* if $v_i$ is the right extremity or otherwise *left* in *dir*. If $A$ can move $H'$ hops to *dir* by the $(t + 2H' + |V^t| - 2)$-th step according to the view, $A$ does so (Figure 2b). Otherwise, $A$ moves $|V^t| - 1 + H'$ hops to $\overline{dir}$ (Figure 2c). Notice that $A$ can decide this condition because $H' \leq H$ and $2H' + |V^t| - 2 \leq S$.

---
**Algorithm 1** $\text{EXPH}(t, v_i)$

---
1: $dir \leftarrow Extremity(t, v_i)$
2: **if** $A$ can move $H'$ hops to $dir$ by the $(t + 2H' + |V^t| - 2)$-th step **then**
3:     Move $H'$ hops to $dir$
4: **else**
5:     Move $|V^t| - 1 + H'$ hops to $\overline{dir}$
6: Wait until the $(t + 2H' + |V^t| - 2)$-th step

---

**Lemma 2.** *Suppose that at the $t$-th step, $A$ exists at the right or left extremity, say $v_i$, of $V^t$ and starts $\text{EXPH}(t, v_i)$. If $2H' + |V^t| - 1 \leq \min(S + 1, n)$, $A$ explores $H'$ nodes by the $t'$-th step (the end of $\text{EXPH}(t, v_i)$) and exists on the right or left extremity of $V^{t'}$ at the $t'$-th step where $t' = t + 2H' + |V^t| - 2$.*

**Proof.** Without loss of generality, we assume $v_i$ is the right extremity of $V^t$. Let $m = |V^t|$, $E_r = \{e_i, e_{i+1}, \ldots, e_{i+H'-2}, e_{i+H'-1}\}$, and $E_l = \{e_{i-H'-m+1}, e_{i-H'-m+2}, \ldots, e_{i-2}, e_{i-1}\}$. Note that since $|E_r| + |E_l| = 2H' + m - 1$ and $2H' + m - 1 \leq n$, $E_r \cap E_l = \emptyset$.

Now, consider the move of $A$. Since $2H' + m - 1 \leq S + 1$, i.e., $2H' + m - 2 \leq S$, $A$ can see whether it can move $H'$ hops to right by the $(t + 2H' + m - 2)$-th step or not.

If $A$ can, $A$ moves $H'$ hops to right and thus the lemma holds.

Otherwise, $A$ can move at most $H' - 1$ hops to right by the $(t + 2H' + m - 2)$-th step, which means during the $2H' + m - 2$ steps, there exists at least $2H' + m - 2 - (H' - 1) = H' + m - 1$ steps at each of which one of the links in $E_r$ is missing. Since at most one link is missing at each step and $E_r \cap E_l = \emptyset$, every link in $E_l$ exists at each of the $H' + m - 1$ steps. Thus, $A$ succeeds to move $H' + m - 1$ hops to left and the lemma holds. □

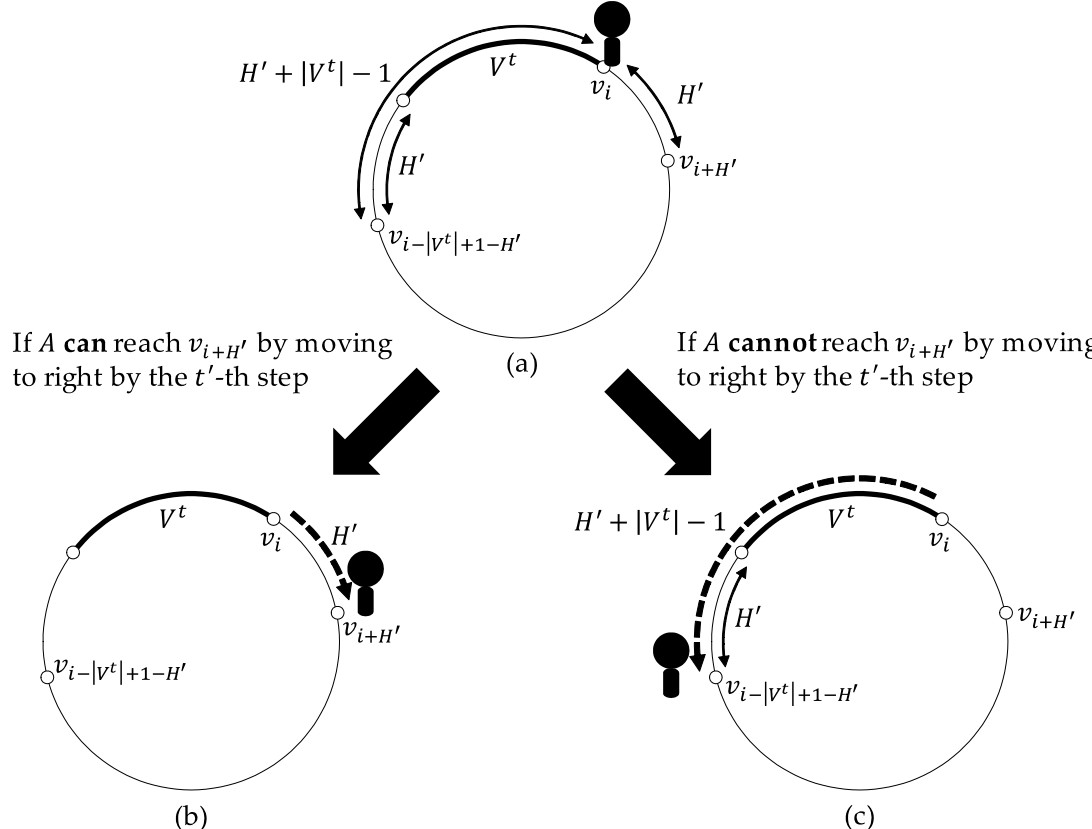

**Figure 2.** The moves of $A$ by EXPH$(t, v_i)$ where $t' = t + 2H' + |V^t| - 2$ in the case where $v_i$ is the right extremity of $V^t$. (**a**) At the start of EXPH$(t, v_i)$, $A$ exists on $v_i$. (**b**) If $A$ can reach $v_{i'+H}$ by moving to right by the $t'$-th step, $A$ moves to right and reaches $v_{i+H'}$ by the $t'$-th step. (**c**) Otherwise, $A$ moves to left and reaches $v_{i-|V^t|+1-H'}$ by the $t'$-th step.

**EXPONE.** EXPONE$(t, v_i)$ described in Algorithm 2 is an algorithm by which $A$ explores at least one node or completes the exploration when $A$ starts EXPONE$(t, v_i)$ from $v_i$ at the $t$-th step under the assumption that $v_i$ is the right or left extremity of $V^t$. Note that in the following, when $A$ executes EXPONE$(t, v_i)$, $A$ is always on the right or left extremity of $V^t$.

When starting the algorithm, $A$ first sees if $v_i$ is the right extremity or the left one and stores the direction in *dir*. Variables $i'$ and $i''$ are used to remember the *dir* neighbor of $v_i$ and the *dir* incident edge of $v_i$ respectively, e.g., $i' = i + 1$ (resp., $i' = i - 1$) if *dir* is *right* (resp., *left*). Then, $A$ stores $\max(n - H, \lceil n/2 \rceil)$ to $S'$ which is not larger than $S$ and is used instead of $S$ in the algorithm.

After that, if $e_{i''}$ appears by the $(t + S' - 1)$-th step, $A$ waits at $v_i$ until $e_{i''}$ appears and moves to $v_{i'}$ when $e_{i''}$ appears. Otherwise, for each $0 \le d \le H - 1$, $A$ moves one hop to $\overline{dir}$ at the $(t + d)$-th step if $e_{i''}$ is missing at the $(t + S' - 1 + d)$-th step in its view (Figure 3a). If $A$ sees $e_{i''}$ appear at the $(t + S' - 1 + d)$-th step in its view at the $(t + d)$-th step, then $A$ starts to move *dir* from the $(t + d)$-th step, returns to $v_i$, waits at $v_i$ until $e_{i''}$ appears, and reaches $v_{i'}$ through $e_{i''}$ (Figure 3b). When $d$ reaches $H$, i.e., $A$ moves $H$ hops to $\overline{dir}$ and $e_{i''}$ is no longer included in the view of $A$, $A$ starts to keep moving to $\overline{dir}$ until reaching $v_{i'}$ and the exploration finishes when reaching $v_{i'}$ (Figure 3c).

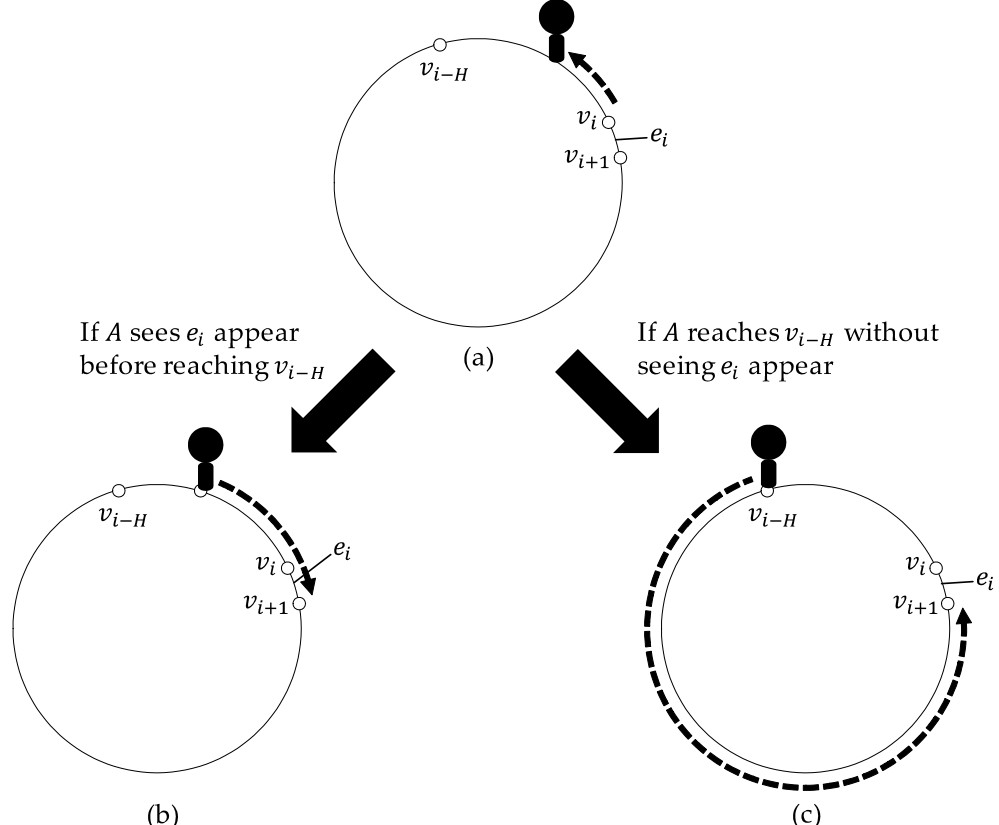

**Figure 3.** The moves of $A$ by ExpOne$(t, v_i)$ in the case where $v_i$ is the right extremity. (**a**) Unless $A$ sees $e_i$ appear, $A$ moves to left. (**b**) If $A$ sees $e_i$ appear before reaching $v_{i-H}$, $A$ starts to move to right and reaches $v_{i+1}$. (**c**) If $A$ reaches $v_{i-H}$ without seeing $e_i$ appear, $A$ keeps moving to left until reaching $v_{i+1}$ and finishes the exploration.

---

**Algorithm 2** ExpOne$(t, v_i)$

---

1: $dir \leftarrow Extremity(t, v_i)$
2: **if** $dir$ is *right* **then**
3:    $i' \leftarrow i + 1, i'' \leftarrow i$
4: **else**
5:    $i' \leftarrow i - 1, i'' \leftarrow i - 1$
6: $d \leftarrow 0$
7: $S' \leftarrow \max(n - H, \lceil n/2 \rceil)$
8: **while** $(d < H)$ **do**
9:    **if** $e_{i''}$ is always missing until the $(t + d + S' - 1)$-th step **then**
10:      Move one hop to $\overline{dir}$
11:      $d \leftarrow d + 1$
12:    **else**
13:      Move $d$ hops to $dir$ (reach $v_i$)
14:      Wait for $e_{i''}$ to appear and pass through $e_{i''}$ as soon as it appears
15:      Exit from the while loop
16: **if** $(d \geq H)$ **then**
17:    Move $n - H - 1$ hops to $\overline{dir}$ (reach $v_{i'}$)
18: Wait until the $(t + n)$-th step

---

**Lemma 3.** *Suppose that at the $t$-th step, $A$ exists at the right (resp., left) extremity, say $v_i$, of $V^t$ and starts* ExpOne$(t, v_i)$. *Then, $A$ completes the exploration or reaches $v_{i+1}$ (resp., $v_{i-1}$) by the $(t + n)$-th step (the end*

of EXPONE$(t, v_i)$). In addition to that, $A$ exists on the right or left extremity of $V^{t+n}$ at the $(t + n)$-th step when the exploration has not been completed.

**Proof.** Without loss of generality, we assume $v_i$ is the right extremity of $V^t$. As in Algorithm 2, let $S' = \max(n - H, \lceil n/2 \rceil)$. We first show the lemma for the case $e_i$ appears by the $(t + d + S' - 1)$-th step in $A$'s view at the $(t + d)$-th step for $0 \le d \le H - 1$.

For $d = 0$, $A$ can clearly reach $v_{i+1}$ by the $(t + S' - 1)$-th step.

For $1 \le d \le H - 1$, when $A$ sees $e_i$ appear for the first time at the $(t + d + S' - 1)$-th step in its view at the $(t + d)$-th step, $e_i$ must appear at the $(t + d + S' - 1)$-th step and be missing at the $t'$-th step for $t + d \le t' \le t + d + S' - 2$ by the construction. This means that all the other links than $e_i$ are present at the $t'$-th step $(t + d \le t' \le t + d + S' - 2)$, and thus $A$ can move for $S' - 1$ steps from $v_{i-d}$ to right without interference by missing links until reaching $v_i$.

Since $d \le H - 1$ and $H \le S'$, $A$ always reaches $v_i$ by the $(t + d + S' - 1)$-th step at which $e_i$ appears. Then, $A$ reaches $v_{i+1}$ as soon as $e_i$ appears. Since $A$ moves at most $H - 1$ hops to left, $e_i$ appears $S'$ steps after $A$ starts to move to right, and $H - 1 + S' \le n$ from $S' = \max(n - H, \lceil n/2 \rceil)$, $A$ reaches $v_{i+1}$ through $e_i$ by the $(t + n)$-th step.

We then show for the other case, i.e., $A$ reaches $v_{i-H}$ at the $(t + H)$-th step. When this happens, $e_i$ must be deleted for at least $S' - 1$ steps from the $(t + H)$-th step and all the other links than $e_i$ are present in the $S' - 1$ steps. Thus, $A$ can move for $S' - 1 \ge n - H - 1$ steps from $v_{i-H}$ to left without interference by missing links until reaching $v_{i+1}$ since $S' = \max(n - H, \lceil n/2 \rceil)$. Since $H + n - H - 1 = n - 1$, $A$ reaches $v_{i+1}$ after $n - H - 1$ steps, i.e., at the $(t + n - 1)$-th step, and the exploration is completed at the same time. □

**Exploration algorithm.** Algorithm 3 describes the exploration algorithm. Let $S'' = \min(S, n - 1)$. The algorithm repeats EXPH$(t, v_i)$ for $\lfloor (S'' + 1 - H')/H' \rfloor$ times (lines 2-6) and EXPONE$(t, v_i)$ for $n - H' \lfloor (S'' + 1 - H')/H' \rfloor - 1$ times (lines 7-13). We call the part repeating EXPH$(t, v_i)$ (lines 2-6) *the first part* and the part repeating EXPONE$(t, v_i)$ *the second part* (lines 7-13). In the first part, $H' \lfloor (S'' + 1 - H')/H' \rfloor + 1$ nodes are explored and, in the second part, the remaining $n - H' \lfloor (S'' + 1 - H')/H' \rfloor - 1$ nodes are explored.

---

**Algorithm 3** Exploration algorithm for $H + S \ge n$

---

1:　$S'' \leftarrow \min(S, n - 1)$
2:　$p \leftarrow 1$ //starting the first part
3:　**while** $(p \le \lfloor (S'' + 1 - H')/H' \rfloor)$ **do**
4:　　Let $t$ be the current step and $v_i$ be the current node
5:　　EXPH$(t, v_i)$
6:　　$p \leftarrow p + 1$
7:　$p \leftarrow 1$ //starting the second part
8:　**while** $(p \le n - H' \cdot \lfloor (S'' + 1 - H')/H' \rfloor - 1)$ **do**
9:　　Let $t$ be the current step and $v_i$ be the current node
10:　　EXPONE$(t, v_i)$
11:　　**if** Exploration is completed **then**
12:　　　Exit from the while loop
13:　　$p \leftarrow p + 1$

---

**Theorem 1.** *For $H + S \ge n$ and $S \ge \lceil n/2 \rceil$, the exploration time of 1-interval connected rings by a single agent with the $(H, S)$ view is upper-bounded by $O(n^2/H + nH)$ if $2H' - 1 \le S$ or otherwise it is upper-bounded by $O(n^2)$.*

**Proof.** It suffices to show that $A$ with the $(H, S)$ view completes exploration within $O(n^2/H + nH)$ steps if $2H' - 1 \leq S$ or otherwise $O(n^2)$ steps by executing Algorithm 3 when $H + S \geq n$ and $S \geq \lceil n/2 \rceil$.

We first consider the case where $2H' - 1 \leq S$. In this case, since $\lfloor (S'' + 1 - H')/H' \rfloor \geq 1$, the first part is executed at least once. Consider the first part. Let $t_p$ be the step when $A$ starts the $p$-th ExpH$(t, v_i)$.

We show by induction that for $1 \leq p \leq \lfloor (S'' + 1 - H')/H' \rfloor$, $|V^{t_p}| = (p-1)H' + 1$ and $A$ explores $H'$ nodes by ExpH$(t_p, v_i)$.

For the base case, i.e., $p = 1$, $|V^{t_1}|$ is clearly $1 = (p-1)H' + 1$. This leads to that $2H' + |V^t| - 1 = 2H' \leq \min(n, S + 1)$. Then, by Lemma 2, $A$ explores $H'$ nodes by ExpH$(t_1, v_i)$.

Now, for $k \leq \lfloor (S'' + 1 - H')/H' \rfloor - 1$, assume that $|V^{t_k}| = (k-1)H' + 1$ and $A$ explores $H'$ nodes by ExpH$(t_k, v_i)$. Then, clearly $|V^{t_{k+1}}| = (k-1)H' + 1 + H' = kH' + 1$. Since $k \leq \lfloor (S'' + 1 - H')/H' \rfloor - 1$, $2H' + |V^{t_{k+1}}| - 1 < n$ and $2H' + |V^{t_{k+1}}| - 1 < S + 1$. Thus, $A$ explores $H'$ nodes by ExpH$(t_{k+1}, v_i)$.

By Lemma 2, $S'' = O(n)$, and $H' = \Theta(H)$, the exploration time of the first part is

$$\sum_{p=1}^{\lfloor (S''+1-H')/H' \rfloor} (2H' + |V^{t_p}| - 2) = \sum_{p=1}^{\lfloor (S''+1-H')/H' \rfloor} ((p+1)H' - 1) = O(n^2/H).$$

We then consider the second part. By Lemma 2, $A$ exists at the right or left extremity of $V^t$ and $|\overline{V^t}| = n - H' \lfloor (S'' + 1 - H')/H' \rfloor - 1 = O(H)$ at the start of the second part. Thus, since $A$ explores one node within $n$ steps by Lemma 3, the exploration time of the second part is $O(nH)$.

As a result, the exploration time of Algorithm 3 is $O(n^2/H + nH)$ when $2H' - 1 \leq S$.

When $2H' - 1 > S$, the first part is never executed and then the number of remaining nodes at the start of the second part is $n - 1$. Thus, in this case, the exploration time of Algorithm 3 is $O(n^2)$. □

From Lemma 1 and Theorem 1, the following theorem holds.

**Theorem 2.** *If and only if $H + S \geq n$ and $S \geq \lceil n/2 \rceil$, a single agent with the $(H, S)$ view can explore of 1-interval connected rings within finite time steps.*

## 5. Upper Bound of Exploration Time for $S \geq N - 1$

In this section, we consider the upper bound of the exploration time when $S \geq n - 1$. We show that the upper bound of the exploration time is reduced to $O(n^2/H + n \log H)$ in this case by giving an exploration algorithm.

We first introduce a new operation ExpHalf$(t, v_i)$ that is used as a building block to construct the exploration algorithm.

**ExpHalf.** ExpHalf$(t, v_i)$ described in Algorithm 4 is an algorithm by which $A$ explores $\lceil |\overline{V^t}|/2 \rceil$ nodes when $A$ starts ExpHalf$(t, v_i)$ from $v_i$ at the $t$-th step under the assumption that $v_i$ is the right or left extremity of $V^t$, $|\overline{V^t}| \leq 2H$, and $S \geq n - 1$. Note that in the following, when $A$ executes ExpHalf$(t, v_i)$, $A$ is always on the right or left extremity of $V^t$.

When starting the algorithm, $A$ first sees if $v_i$ is the right extremity or the left one and stores *right* if $v_i$ is the right extremity or otherwise *left* in *dir*. If $A$ can move $\lceil |\overline{V^t}|/2 \rceil$ hops to *dir* by the $(t + n - 1)$-th step according to the view, $A$ does so (Figure 4b). Otherwise, $A$ moves $n - |\overline{V^t}|/2$ hops to $\overline{dir}$ (Figure 4c).

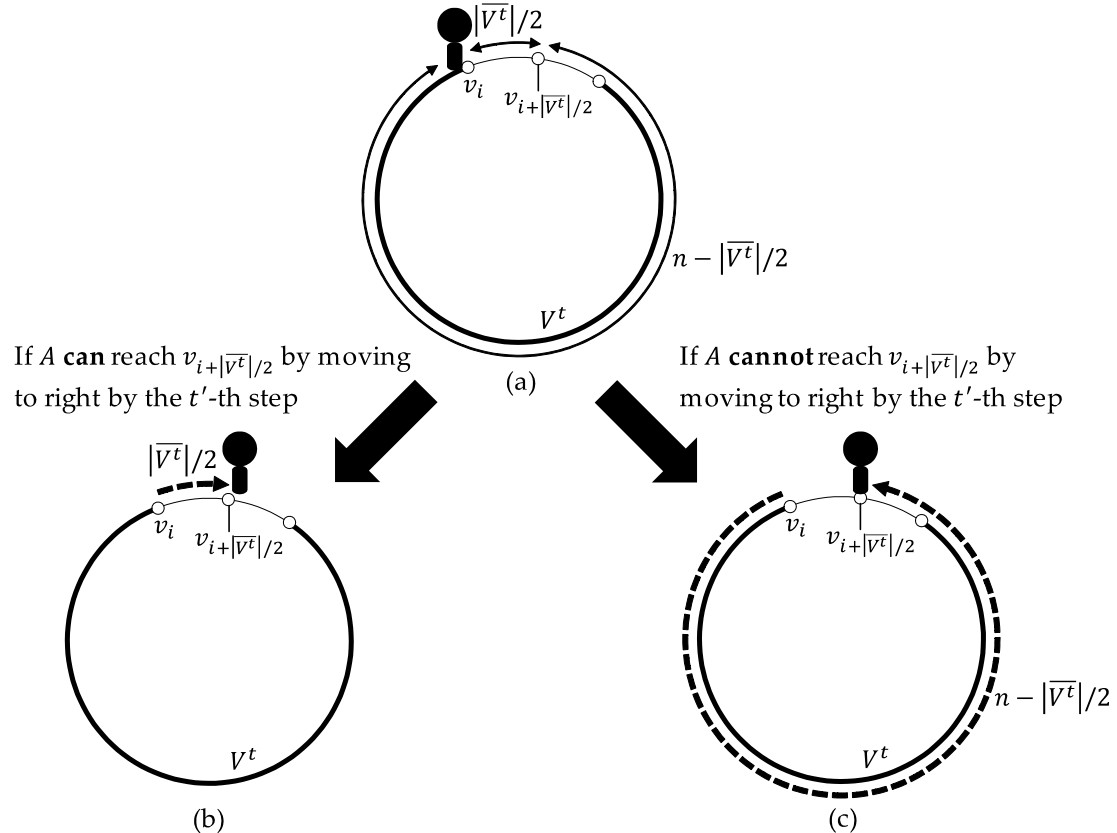

**Figure 4.** The moves of $A$ by EXPHALF$(t, v_i)$ where $t' = t + n - 1$ in the case where $v_i$ is the right extremity of $V^t$. (**a**) At the start of EXPHALF$(t, v_i)$, $A$ exists on $v_i$. (**b**) If $A$ can reach $v_{i+|\overline{V^t}|/2}$ by moving to right by the $t'$-th step, $A$ moves to right and reaches $v_{i+|\overline{V^t}|/2}$ by the $t'$-th step. (**c**) Otherwise, $A$ moves to left and reaches $v_{i+|\overline{V^t}|/2}$ by the $t'$-th step.

---

**Algorithm 4** EXPHALF$(t, v_i)$

---

1: $dir \leftarrow Extremity(t, v_i)$
2: **if** $A$ can move $\lceil|\overline{V^t}|/2\rceil$ hops to $dir$ by the $(t+n-1)$-th step **then**
3: 　　Move $\lceil|\overline{V^t}|/2\rceil$ hops to $dir$
4: **else**
5: 　　Move $n - \lceil|\overline{V^t}|/2\rceil$ hops to $\overline{dir}$
6: Wait until the $(t + n - 1)$-th step

---

**Lemma 4.** *Suppose that at the $t$-th step, $A$ exists at the right or left extremity, say $v_i$, of $V^t$ and starts* EXPHALF$(t, v_i)$. *If $|\overline{V^t}| \le 2H$ and $S \ge n - 1$, $A$ can explore at least $\lceil\overline{V^t}/2\rceil$ nodes by the $t'$-th step (the end of* EXPHALF$(t, v_i)$*) and exists on the right or left extremity of $V^{t'}$ at the $t'$-th step where $t' = t + n - 1$.*

**Proof.** Without loss of generality, we assume $v_i$ is the right extremity of $V^t$. Let $m = |\overline{V^t}|$, $E_r = \{e_i, e_{i+1}, \ldots, e_{i+\lceil m/2\rceil - 1}, e_{i+\lceil m/2\rceil}\}$, and $E_l = \{e_{i+\lceil m/2\rceil + 1}, e_{i+\lceil m/2\rceil + 2}, \ldots, e_{i-1}\}$.

Now, consider the move of $A$. Since $S \ge n - 1$ and $m \le 2H$, $A$ can see whether it can move $\lceil m/2\rceil$ hops to right by the $(t + n - 1)$-th step or not.

If $A$ can move $\lceil m/2\rceil$ hops, $A$ moves $\lceil m/2\rceil$ hops to right and thus the lemma holds.

Otherwise, $A$ can move at most $\lceil m/2\rceil - 1$ hops to right by the $(t + n - 1)$-th step, which means during the $n - 1$ steps, there exist at least $n - 1 - (\lceil m/2\rceil - 1) = n - \lceil m/2\rceil$ steps at each of which one of the links in $E_r$ is missing. Since at most one link is missing at each step and $E_r \cap E_l = \emptyset$, every link

in $E_l$ exists at each of the $n - \lceil m/2 \rceil$ steps. By this and $|E_l| = n - \lceil m/2 \rceil$, $A$ succeeds to reach $v_{i+\lceil m/2 \rceil}$ by moving to left, which means at least $\lceil m/2 \rceil$ nodes are explored. □

**Exploration algorithm.** Algorithm 5 describes the exploration algorithm. The algorithm repeats $\text{EXPH}(t, v_i)$ for $\lfloor (n - H')/H' \rfloor$ times (lines 1–5) and $\text{EXPHALF}(t, v_i)$ for $\lceil \log(n - H' \lfloor (n - H')/H' \rfloor - 1) \rceil$ times (lines 6–10). We call the part repeating $\text{EXPH}(t, v_i)$ (lines 1-5) *the first part* and the part repeating $\text{EXPHALF}(t, v_i)$ *the second part* (lines 6–10). In the first part, $H' \lfloor (n - H')/H' \rfloor + 1$ nodes are explored and, in the second part, the remaining $n - H' \lfloor (n - H')/H' \rfloor - 1$ nodes are explored.

---

**Algorithm 5** Exploration algorithm for $S \geq n - 1$

---

　1: $p \leftarrow 1$ //starting the first part
　2: **while** $(p \leq \lfloor (n - H')/H' \rfloor)$ **do**
　3: 　　Let $t$ be the current step and $v_i$ be the current node
　4: 　　$\text{EXPH}(t, v_i)$
　5: 　　$p \leftarrow p + 1$
　6: $p \leftarrow 1$ //starting the second part
　7: **while** $(p \leq \lceil \log(n - H' \lfloor (n - H')/H' \rfloor - 1) \rceil)$ **do**
　8: 　　Let $t$ be the current step and $v_i$ be the current node
　9: 　　$\text{EXPHALF}(t, v_i)$
　10: 　　$p \leftarrow p + 1$

---

**Theorem 3.** *For $S \geq n - 1$, the exploration time of 1-interval connected rings by a single agent with the $(H, S)$ view is upper-bounded by $O(n^2/H + n \log H)$.*

**Proof.** It suffices to show that $A$ completes exploration within $O(n^2/H + n \log H)$ steps by Algorithm 5 when $S \geq n - 1$. It is proven that the total exploration time of the first part is $O(n^2/H)$ and that of the second part is $O(n \log H)$.

We first consider the first part. Note that, since $2H' \leq n$, $1 \leq \lfloor (n - H')/H' \rfloor$ and thus the first part is always executed at least once. Let $t_p$ be the step when $A$ starts the $p$-th $\text{EXPH}(t, v_i)$. We can show that for $1 \leq p \leq \lfloor (n - H')/H' \rfloor$, $A$ can explore $H'$ nodes by $\text{EXPH}(t_p, v_i)$ by induction and the exploration time of the first part is $O(n^2/H)$ as in the proof of Lemma 2.

We then consider the second part. By Lemma 2, $A$ exists at the right or left extremity of $V^t$ and $|\overline{V^t}| = n - H' \lfloor (n - H')/H' \rfloor - 1 \leq 2H'$ at the start of the second part. Thus, since $A$ explores a half of $\overline{V^t}$ within $n - 1$ steps by Lemma 4, the exploration time of the second part is $O(n \log H)$. As a result, the exploration time of Algorithm 3 is $O(n^2/H + n \log H)$. □

## 6. Lower Bound of Exploration Time

A lower bound of the exploration time for any $S$ is given in this section. The following theorem holds.

**Theorem 4.** *The exploration time of 1-interval connected rings by a single agent with the $(H, S)$ view is lower-bounded by $\Omega(n^2/H)$.*

**Proof.** We first show that, provided that $A$ is at the right or left extremity, say $v_i$, of $V^t$ at the $t$-th step where $|V^t| \leq n - 2H + 1$, it takes at least $|V^t| + H - 1$ steps for $A$ to explore $H$ nodes from the circumstance under the following link scheduling: $e_{i+H-1}$ (resp., $e_{i-H}$) is deleted until the $(t + |V^t| + H - 1)$-th step if $v_i$ is the right (resp., left) extremity of $V^t$. Without loss of generality, we assume that $v_i$ is the right extremity of $V^t$ in the following. Figure 5 depicts the situation.

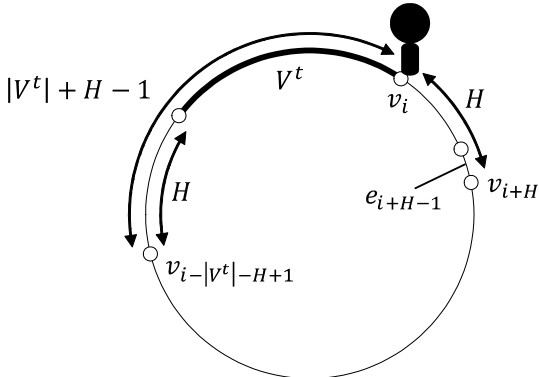

**Figure 5.** The situation where $A$ exists on $v_i$ at the $t$-th step ($v_i$ is the right extremity of $V^t$). The adversary deletes $e_{i+H-1}$ until the $(t+|V^t|+H-1)$-th step in this situation.

Assume for contradiction that $A$ explores the $H$ nodes within $|V^t|+H-1$ steps under the scheduling. Since $e_{i+H-1}$ is missing until the $(t+|V^t|+H-1)$-th step, $A$ never reaches $v_{i+H}$. Therefore, $A$ must explore at least one node on the left side of $V^t$. This and exploring $H$ nodes take at least $|V^t|+H-1$ steps; a contradiction.

Now, apply the above claim from the first step repeatedly. When applying the claim for the $p$-th time, $|V^t| = (p-1)H+1$ and then it takes $|V^t|+H-1 = pH$ steps. Note that we can apply the claim while $(p-1)H+1 \leq n-2H+1$, i.e., $p \leq \lfloor (n-H)/H \rfloor$. We then derive the lower bound of the exploration time, $\sum_{p=1}^{\lfloor (n-H)/H \rfloor} pH = \Omega(n^2/H)$. $\quad\square$

## 7. Discussion

In this paper, we studied the exploration problem on dynamic networks with its partial information, where we focused on 1-interval connected rings as a first step. In this section, we discuss what happens when we consider other connectivity and/or general graphs.

When considering 1-interval connected rings, we yields the restriction that at most one link is missing at each step. By this restriction, an agent gets to know that all the links outside its view exist when a link in its view is missing and can make the action plan to visit an unvisited node using the information. It is interesting to investigate such conditions on the space and the time of a view ($H$ and $S$ in this paper) for more general graphs under some assumptions on the temporal connectivity and/or more general graphs. On the other hand, even under the assumption of 1-interval connectivity and/or the restriction on the number of missing links at each step, an agent cannot necessarily get the whole information of the temporal topology, which may prevent the agent from making the action plan to visit an unvisited node and makes the exploration problem more challenging.

We also conjecture that the space and the time of a view which are necessary and sufficient for an agent to explore depend on temporal diameter. Intuitively, temporal diameter is the maximum duration of the foremost path (the path with the least duration from a node to another node departing at specified time) in a dynamic network (see e.g., Section 4.6 of [1] for a formal definition). The fact that the temporal diameter of a 1-interval connected graph with $n$ nodes is at most $n-1$ fits a possibility result of this paper, i.e., $H+S \leq n$. To investigate the relation of temporal distance and the power of a view is one of the intriguing research directions.

## 8. Conclusions

In this paper, we introduced the $(H,S)$ view which can be used to model some situations where an agent (or robot) can partly see their nearby environment or can predict the near-future changes of the environment. To the best of our knowledge, this is the first work considering such a model. For a single agent with the $(H,S)$ view, we studied the exploration of 1-interval connected rings. We give some fundamental results, i.e., impossibility of the exploration for $H+S < n$ or $S < \lceil n/2 \rceil$,

possibility of the exploration for $H + S \geq n$ and $S \geq \lceil n/2 \rceil$, and upper bounds and a lower bound of the exploration time for some cases.

**Author Contributions:** Conceptualization, T.G.; Funding acquisition, Y.S., F.O. and T.M.; Supervision, Y.S., F.O. and T.M.; Writing–original draft, T.G.; Writing–review & editing, Y.S., F.O. and T.M. All authors have read and agreed to the published version of the manuscript.

**Funding:** This work was supported by JSPS KAKENHI Grant Numbers 17K19977, 18K11167, 20H04140 and 19H04085 and JST SICORP Grant Numbers JPMJSC1606 and JPMJSC1806.

**Conflicts of Interest:** The authors declare no conflict of interest. The funders had no role in the design of the study; in the collection, analyses, or interpretation of data; in the writing of the manuscript, or in the decision to publish the results.

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
