# Peer review of "Dynamic Ring Exploration with (H,S) Viewâ€"

_algorithms, doi:10.3390/a13060141_

Round 1
Reviewer 1 Report
This paper presents studies of 1-interval connected rings by a single agent with partial information about network changes. I think the paper is well organized, however the mathematical presentation needs to be improved. For example adding more flowchart or figures could help dramatically. Also, the proofs are very hard to follow since the author tends to put many things in one paragraph. I think putting equations line by line or adding flowchart will also help a lot.Author Response
Please see the attachment.

Reviewer 2 Report
This paper investigates exploration of dynamic network by a single mobile agent relying on partial information about network changes instead of having complete information or with no information about network changes. Authors proposes and proves the amount of necessary and sufficient conditions to explore 1-interval connected rings. Based on some assumptions, the paper proposes and proves the upper and lower bounds of the exploration time.
The paper investigated one a very interesting and important problem in the area. The structure of the paper, presentation, and flow of discussion are good. The formulation and modelling of the problem and the logic of the paper is interesting. Overall, I propose to accept the paper with no changes required.
Author Response
Reviewer 2 proposed “to accept the paper with no changes required”, and thus we do not have a response for his/her comments.
Reviewer 3 Report
This paper considers problem of exploring 1-interval connected rings by a single agent with partially known network changes. The authors show the prerequisites for the exploration. They further investigate both upper and lower bounds of the exploration time.
The overall presentation of this paper is well with a proper organization. However, the exploration of ring topology could be limited for a wireless network. The authors may provide some practical considerations for motivating their research.
The authors could comprehensively discuss the applications of their research in both sparse and dense networks rather than simply mention them in the conclusion.
